# Repression of *CADM1* transcription by HPV type 18 is mediated by three-dimensional rearrangement of promoter-enhancer interactions

**Karen Campos-León**[1], **Jack Ferguson**[1¤], **Thomas Günther**[2], **C. David Wood**[3],
**Steven W. Wingett**[4], **Selin Pekel**[1], **Christy S. Varghese**[1], **Leanne S. Jones**[1],
**Joanne D. Stockton**[1], **Csilla Várnai**[1], **Michelle J. West**[3], **Andrew Beggs**[1],
**Adam Grundhoff**[2], **Boris Noyvert**[1,5], **Sally Roberts**[1], **Joanna L. Parish**[1,6]*

1 Department of Cancer and Genomic Sciences, College of Medicine and Health, University of Birmingham, Birmingham, United Kingdom, 2 Leibniz Institute of Virology, Hamburg, Germany, 3 School of Life Sciences, University of Sussex, Brighton, United Kingdom, 4 The Babraham Institute, Babraham Research Campus, Cambridge, United Kingdom, 5 Birmingham CRUK Centre, University of Birmingham, Birmingham, United Kingdom, 6 National Institute of Health Research, Biomedical Research Centre, University of Birmingham, Birmingham, United Kingdom

¤ Current address: Department of Microbes, Infections and Microbiomes, College of Medicine and Health, University of Birmingham, Birmingham, United Kingdom
* j.l.parish@bham.ac.uk

## Abstract

Upon infection, human papillomavirus (HPV) manipulates host cell gene expression to create an environment that is supportive of a productive and persistent infection. The virus-induced changes to the host cell's transcriptome are thought to contribute to carcinogenesis. Here, we show by RNA-sequencing that oncogenic HPV18 episome replication in primary human foreskin keratinocytes (HFKs) drives host transcriptional changes that are consistent between multiple HFK donors. We have previously shown that HPV18 recruits the host protein CTCF to viral episomes to control the differentiation-dependent viral transcriptional programme. Since CTCF is an important regulator of host cell transcription via coordination of epigenetic boundaries and long-range chromosomal interactions, we hypothesised that HPV18 may also manipulate CTCF to contribute to host transcription reprogramming. Analysis of CTCF binding in the host cell genome by ChIP-Seq revealed that while the total number of CTCF binding sites is not altered by the virus, there are a sub-set of CTCF binding sites that are either enriched or depleted of CTCF. Many of these altered sites are clustered within regulatory elements of differentially expressed genes, including the tumour suppressor gene cell adhesion molecule 1 (*CADM1*), which supresses epithelial cell growth and invasion. We show that HPV18 establishment results in reduced CTCF binding at the *CADM1* promoter and upstream enhancer. Loss of CTCF binding is coincident with epigenetic repression of *CADM1,* in the absence of CpG hypermethylation, while adjacent genes including the transcriptional regulator *ZBTB16* are activated. These data indicate that the *CADM1* locus is subject to topological rearrangement following HPV18 establishment. We tested this hypothesis using 4C-Seq (circular chromosome confirmation capture-sequencing) and show that HPV18 establishment causes

provided the original author and source are credited.

**Data availability statement:** All next generation sequencing data sets are available from the European Genome-Phenome Archive by request (EGA; https://ega-archive.org/data-sets/EGAD50000001135). The data has been uploaded to the EGA due to potentially identifying information, data access requirements are sent through EGA to the data access committee administered by the Director of Genomics Birmingham.

**Funding:** This work was funded by grants from the Medical Research Council, UK (https://www.ukri.org/councils/mrc/) awarded to JLP (MR/R022011/1, MR/T015985/1 and MR/N023498/1). JLP research group is also supported by the NIHR Biomedical Research Centre (https://www.birmingham-brc.nihr.ac.uk/) awarded to the University of Birmingham. Work from BN was funded through the Cancer Research UK (https://www.cancerresearchuk.org/funding-for-researchers) Birmingham Centre award C17422/A25154. Work by CDW was funded by Blood Cancer UK (https://bloodcancer.org.uk/) grant 20003 to MJW. The funders had no role in study design, data collection and analysis, decision to publish, or preparation of the manuscript.

**Competing interests:** The authors have declared that no competing interests exist.

a loss of long-range chromosomal interactions between the *CADM1* transcriptional start site and the upstream transcriptional enhancer. These data show that HPV18 manipulates host cell promoter-enhancer interactions to drive transcriptional reprogramming that may contribute to HPV-induced disease progression.

## Author summary

Infection with oncogenic HPV is the cause of numerous cancer types, which generally arise after persistent HPV infection. Upon infection, HPV alters the gene expression profile of infected cells to facilitate virus replication and persistence. Multiple mechanisms of HPV-induced host cell reprogramming have been previously suggested. Here, we show that HPV infection induces rearrangement of specific genomic loci by altering the chromatin binding of the host cell protein CTCF, an important regulator of chromatin architecture. Loss of CTCF binding to a cluster of binding sites at the *CADM1* locus on chromosome 11 is coincident with epigenetic reprogramming and disruption of long-range chromatin interactions, resulting in transcriptional repression of *CADM1*. Our data show that repression of *CADM1* is an early event in HPV-driven disease, preceding hypermethylation of the *CADM1* transcriptional promoter that is frequently observed in HPV-driven cancers, demonstrating a novel mechanism of HPV-induced host cell transcriptional reprogramming.

## Introduction

Oncogenic human papillomavirus (HPV) infection is the cause of most cervical cancers and a significant proportion of other anogenital and oropharyngeal cancers [1]. HPVs exclusively infect the undifferentiated basal keratinocytes of squamous epithelia and gain access to these cells via micro-abrasions [2]. Most infections are resolved by host immune activation but in some cases, the virus avoids immune detection and establishes a persistent infection, a strong risk factor for cancer development. Virus-mediated transcriptional manipulation of the host is an important facet in the maintenance of persistent oncogenic HPV infection. This family of viruses have evolved intricate mechanisms to induce transcriptional changes in the host to evade immune detection and support persistent infection. Several studies have identified gene sets that are transcriptionally deregulated following oncogenic HPV establishment in primary keratinocytes including down-regulation of immune defence-related genes and altered expression of genes important for epithelial cell structure and differentiation [3–6]. However, the mechanisms that drive HPV-induced host transcriptional reprogramming and how these changes contribute to cancer development remain unclear.

HPV oncogenes E6 and E7 modulate the activity of numerous histone modifying and chromatin-remodelling enzymes leading to epigenetic regulation of viral and host chromatin (reviewed by [7]). Epigenetic modulation of host chromatin is associated with both HPV infection and HPV-driven carcinogenesis and multiple mechanisms have been proposed. HPV E6 and E7 can modulate the activity of both histone acetyltransferases (HATs) and histone deacetylases (HDACs) resulting in host transcriptional reprogramming [8,9]. In addition, HPV E7 drives a global decrease in H3K27Me3 abundance via induction of KDM6A and KDM6B lysine 27-specific demethylases resulting in host cell transcriptional reprogramming via homeobox gene activation [10,11].

HPV replication in primary keratinocytes also induces hypermethylation of CpG dinucleotides in the host genome, directly impacting gene expression [12,13]. This is at least in part through up-regulation of DNA methyltransferase 1 (DNMT1) via the action of the viral oncoproteins E6 and E7 [14,15]. Longitudinal models of HPV-driven primary cell immortalisation have demonstrated that initial promoter hypermethylation is targeted rather than randomly distributed sites in the host genome [12] and that sequential increases in hypermethylation of specific promoters correlates with disease progression [12,13]. While oncogenic HPV drives a distinct pattern of host transcriptional reprogramming through targeted epigenetic modulation of gene loci [16,17], how these changes are directed is not known. In addition, whether the histone-mediated epigenetic alteration of host transcription observed in models of HPV infection promotes CpG methylation at CpG islands (CGIs) commonly hypermethylated in HPV-driven cancers remains unanswered.

We have previously demonstrated that HPV18 targets the ubiquitously expressed host protein CCCTC-binding factor (CTCF) to attenuate viral gene expression via the establishment of a CTCF-Yin Yang 1 (YY1)-dependent epigenetically repressed chromatin loop between the viral enhancer and the E2 encoding early gene region [18,19]. CTCF is a zinc finger DNA binding protein that binds to tens of thousands of genomic sites to regulate epigenetic boundaries, nucleosome positioning, and long-range chromatin interactions termed topologically associated domains (TADs). CTCF is frequently mutated in numerous cancer types [20] and isolated somatic mutations in CTCF binding sites have been identified in tumours [21,22]. Disruption of CTCF binding has been shown to have profound effects on gene expression within specific gene loci associated with oncogenic transformation [23,24]. While CTCF is an important regulator of HPV gene expression, it is unknown whether HPV infection alters the cellular function of this key transcriptional regulator. Here, using a physiological model of HPV18 episome replication and maintenance in primary human foreskin keratinocytes [19,25,26] we show that HPV establishment alters the binding footprint of CTCF at discrete cellular loci. Disruption of CTCF binding clusters within specific cellular loci including the *CADM1* (cell adhesion molecule 1; a major regulator of cell adhesion, also known as tumour suppressor in lung cancer 1; *TSLC1*) locus correlates with epigenetic rearrangement and differential gene expression within these loci. We show that loss of CTCF binding at the *CADM1* transcription start site (TSS) and upstream enhancer elements results in topological rearrangement of promoter-enhancer interaction and provides a mechanism for *CADM1* transcriptional repression following HPV establishment. In cervical carcinogenesis, *CADM1* silencing by promoter hypermethylation is a frequent event in the progression from high grade cervical dysplasia to cancer. Our data show that repression of *CADM1* expression occurs early in the establishment of cells maintaining HPV18 episomes, not by promoter hypermethylation but via alteration of CTCF-dependent promoter-enhancer interaction and localised epigenetic rearrangement. Repression of CADM1 as a major regulator of cell adhesion may be a key event in virus replication and in HPV-induced carcinogenesis mediated by different virus-induced mechanisms.

## Results

### Establishment of HPV18 in primary keratinocytes induces alteration of the host cell transcriptome

Global gene expression changes induced by HPV18 transfection and establishment in primary HFKs were measured by RNA-sequencing (RNA-Seq) of polyA+ selected mRNA. HFK cultures harvested from six individual donors were transfected with recircularised HPV18 genomes to establish persistent episomal replication. HFK donor cells and matched HPV18

episome-containing lines (HFK-HPV18) were cultured in monolayer on γ-irradiated J2 fibroblasts in serum containing media and the cellular transcriptome was analysed in cells below passage 5 (representing ~ 20–25 population doublings) to ensure episomal replication of HPV18 genomes, as confirmed by Southern blot analysis of each donor (S1A Fig and [27,28]). Although we were able to identify virus-host fusion transcripts in all six donor lines analysed, these were detected at very low abundance (< 0.3% of total fragments mapped to the HPV18 genome; S1 Table), further confirming minimal transcriptionally active viral integrants. In addition, alignment of HPV18 derived transcripts confirmed expression of early transcripts that are predominantly spliced at E6*I and E1^E4 major splice sites and terminate at the early polyA+ signal downstream of E5, typical of episomal HPV transcription (S1B Fig).

Principal component analysis (PCA) of regularized log (rlog) transformed RNA-Seq data revealed distinct clustering of HFK and HFK-HPV18 samples with a clear difference in PCA1 following HPV18 episome establishment in all six HFK donors (Fig 1A; p = 0.039), and PC2 showing the differences between the replicates. Differential gene expression (DGE) analysis revealed 976 host genes with 2 or more fold change in gene expression and adjusted p < 0.05 following HPV18 establishment in all six donors. Of these genes, 628

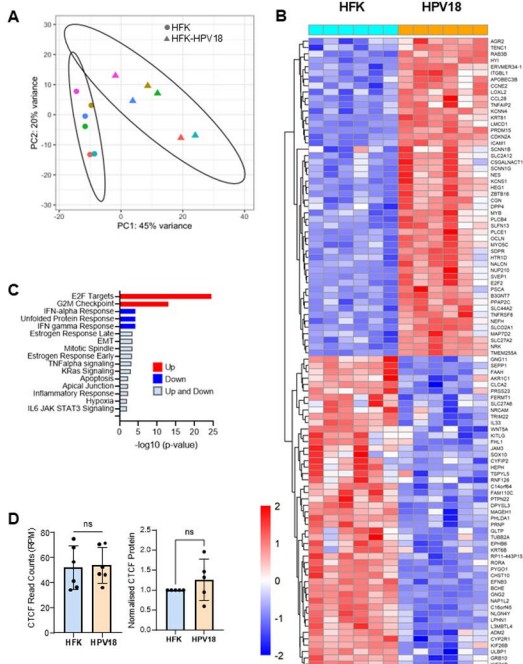

**Fig 1. HPV18 establishment in primary HFKs induces a distinctive pattern of host differential gene expression.** (**A**) Two main principal components of rlog transformed RNA-Seq data from six primary HFK cultures (circles) and isogenic HPV18-genome containing lines (triangles). The six individual donors are shown in different colours. Separation of the 2 groups of samples is shown by the ellipses denoting 95% confidence intervals for each group. Statistical significance of separation was calculated using a permutation test (ClusterSignificance package); p = 0.039. (**B**) Heatmap of unsupervised hierarchical clustering of the top 50 most significantly upregulated and 50 most significantly down regulated genes between HFK (aqua) and HPV18 (orange) samples. Relative low gene expression is shown in blue and high gene expression shown in red as detailed in the legend. (**C**) Gene set enrichment analysis of differential expression of genes following HPV18 establishment using the Molecular Signature Hallmarks database. Directional response in expression of gene sets is indicated and significance of gene set terms (corrected for multiple testing) is given on the x-axis, showing only the terms significant at FDR = 0.01. (**D**) Bar graphs show CTCF read counts (reads per million, RPM) in HFK and HFK-HPV18 cultures taken from RNA-Seq data and densitometry-based analysis of CTCF protein expression levels from 5 donors 5. Data show the mean and standard deviation, *p < 0.05.

were upregulated (S2 Fig and S2 Table) and 348 were downregulated (S2 Fig and S3 Table). Unsupervised hierarchical clustering of the top 50 most up regulated and down regulated genes revealed a remarkably consistent alteration of host cell gene expression in all six HFK donors, suggesting that genes are specifically targeted by HPV18 regardless of host cell genetic background (Fig 1B).

To determine the biological pathways that are transcriptionally altered following HPV18 establishment, we performed gene set enrichment analysis (GSEA) using the Molecular Signatures Database (MSigDB) and Hallmark gene set (http://www.gsea-msigdb.org/). A total of 16 pathways were significantly enriched (FDR < 0.01) including up-regulation of E2F targets and G2M checkpoint and down-regulation of immune response pathways (interferon alpha and gamma responses) and the unfolded protein response. Other notable pathways were both up- and down-regulated including epithelial to mesenchymal transition (EMT), mitotic spindle, apoptosis, immune signalling pathways (TNFα signalling via NFκB and IL-6 JAK-STAT3 signalling) and hypoxia (Fig 1C).

## HPV18 alters CTCF binding site occupancy in the host cell genome

To discover novel mechanisms of HPV18-driven host cell gene expression changes, we opted to identify potential changes to host cell chromatin architecture by mapping differential CTCF binding. CTCF is an important regulator of host cell gene expression, partially through the stabilisation of promoter-enhancer interactions and the insulation of epigenetic boundaries [29], and we therefore hypothesised that alteration of CTCF activity could drive HPV-mediated host gene expression changes. We and others have previously reported that oncogenic HPV establishment resulted in a significant increase in CTCF protein level [30,31] that was not due to transcriptional upregulation of CTCF [30], suggesting that changes to CTCF protein function or turnover could be altered by HPV. To validate this finding in the HFK cultures used in this study, CTCF transcript levels were quantified in the RNA-Seq datasets, which showed no change upon HPV18 establishment in the six HFK donor lines sequenced as previously reported. However, western blot analysis confirmed a small average increase in CTCF protein abundance, but this increase did not reach statistical significance and was not consistent between individual donors (Fig 1D). To determine whether HPV18 replication could induce a change in the distribution of CTCF binding within the human genome and if this might contribute to the altered host cell gene expression profile observed, we analysed CTCF binding distribution by chromatin immunoprecipitation-sequencing (ChIP-Seq) in two independent HFK donor lines (donors 3 and 4). CTCF-bound peaks were identified using Model-based Analysis of ChIP-Seq (MACS) in each HFK donor before and after HPV18 establishment. While no difference in the total number of CTCF-bound peaks was observed following HPV18 establishment (HFK_3 n = 62125; HFK_4 n = 40988; HFK-HPV18_3 n = 59704, HFK-HPV18_4 n = 41105; p = 0.53), differential peak analysis revealed a subset of peaks differentially bound following HPV18 establishment (>log2FC ±0.5, p < 0.005) (Fig 2A and 2B and S4 Table). In total, 398 peaks were significantly reduced in CTCF binding and 504 peaks were significantly increased in the two donors tested. Annotation of all CTCF peaks identified by genomic feature revealed a similar distribution within the human genome as previously described [32] with most peaks within genebody (40%) or intergenic regions (41%). In addition, 14% of peaks were within proximal and distal promoter regions and less than 5% of peaks were within gene deserts (Fig 2C and S4 Table). Interestingly, the distribution of differentially bound peaks in HFK-HPV18 compared to HFK showed a bias towards gene desert regions, particularly in the peaks that were significantly reduced following HPV18 establishment where 19.4% of differentially accessible CTCF peaks were found within gene deserts (Fig 2C).

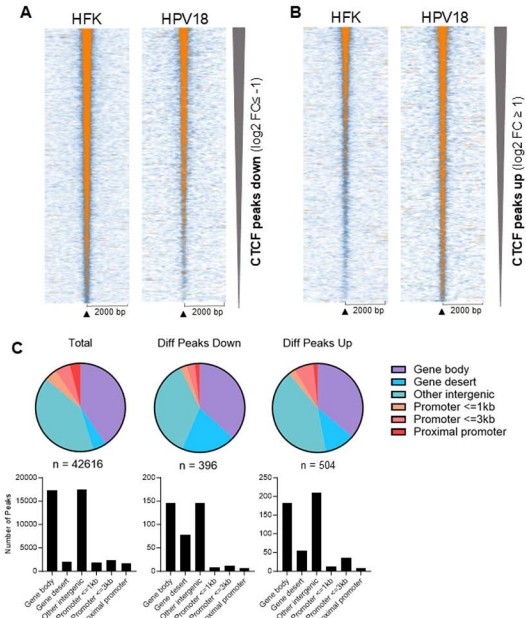

**Fig 2. HPV18 induces differential host CTCF binding in primary HFKs.** CTCF-bound peaks were identified by ChIP-Seq analysis in two independent HFK donors. Significantly enriched peaks were identified using MACs (version 1.4.3) with a p-value cut-off of $1 \times 10^{-5}$. Heatmaps of CTCF peaks that were significantly down- (**A**) or up-regulated (**B**) in HPV positive cells with a log2 fold change cutoff of −1 and +1, respectively. Heatmaps are sorted according to decreasing CTCF ChIP-Seq signal in HFKs. Normalized CTCF signal intensities are illustrated by white/blue to orange color gradient ranging from 0 to 5. (**C**) Pie charts showing the genomic features associated with all CTCF peaks identified (total) and the peaks significantly reduced (Diff Peaks Down) or increased in magnitude (Diff peaks Up). Genomic feature annotations were obtained using the diffReps tool.

To determine whether novel CTCF binding sites were occupied following HPV18 establishment, the CTCF binding peaks identified in our datasets were compared to previously described CTCF binding sites from experimental data (https://remap.univ-amu.fr/target_page/CTCF:9606). This comparison revealed that 99.0% of CTCF peaks identified in our ChIP-Seq experiment had been previously identified. Analysis of the 902 differentially bound peaks in HPV18-HFK compared to HFK revealed that 97.9% of the differential peaks have been previously described. Of the 2.1% of differential peaks not overlapping with the REMAP database, none were located at the *CADM1* locus. These new data indicate that most of the differentially bound peaks in HPV18-HFK compared to HFK are previously annotated *bone fide* CTCF binding sites, rather than novel sites, that are differentially occupied in response to HPV18 replication establishment.

## HPV18-induced alteration of CTCF binding frequently occurs in clusters associated with differential gene expression

To determine whether HPV18-induced differential CTCF binding in the human genome contributes to any of the observed host transcriptional changes, we integrated RNA-Seq analysis of DGE with CTCF ChIP-Seq data. The distance of genes that were altered at least 2-fold (up and down) from the nearest at least 2-fold differentially bound CTCF peak was calculated. This analysis revealed that of the 976 genes that were differentially expressed ($|\log_2 FC| > 1.0$ or $< -1.0$ and adjusted p-value $< 0.05$), 107 (11%) of these genes were within 100Kb of differentially bound CTCF sites (S5 Table). This represents a 1.5-fold enrichment (p = $2.3 \times 10^{-5}$,

Hypergeometric test) compared to the 7.5% (1,688 out of 22,598) of all genes included in the differential expression analysis. When focusing on genes that directly overlapped with differential CTCF binding sites (distance = 0 in S5 Table), we observed a 2.7-fold enrichment, with 38 differentially expressed genes (3.9%) compared to 323 (1.43%) of all genes, yielding a Hypergeometric test p-value of $7.3 \times 10^{-9}$.

Interestingly, there appeared to be close clustering of multiple differential CTCF peaks within distinct genomic loci, suggesting that these loci have undergone epigenetic and/or topological rearrangement following HPV18 episome establishment. To identify all loci with multiple differentially bound CTCF peaks and corresponding differential expression we used the bedtools cluster function to identify differential CTCF peak clusters based on the distance between differentially bound peaks. In total, eight genomic loci were identified which contained 12 or more differentially bound CTCF sites. Seven of these loci contained at least one significantly differentially expressed gene (Tables 1 and S6 and S4 Fig). This analysis identified a region on chromosome 11 which contains a cluster of 20 differentially bound CTCF binding sites (Fig 3A). As this was the highest number of differentially bound CTCF peaks within a distinct genomic locus following HPV18 establishment, we decided to focus our investigation and downstream experiments on this genomic locus. The cell adhesion molecule 1 (*CADM1)* and zinc finger and BTB domain 16 (*ZBTB16*; also known as promyelocytic leukaemia zinc finger, *PLZF*) genes are located within this locus. Notably the expression of *CADM1* is significantly reduced in our RNA-Seq data (Fig 3B and S3 Table; Log2FC = −3.1, p = 2.83e-06), which was validated by qRT-PCR using three independent primer sets (Fig 3B) and CADM1 protein expression was significantly reduced in HFK-HPV18 in comparison to isogenic HFK cultures (Fig 3C). Conversely, *ZBTB16* mRNA was significantly increased in both our RNA-Seq data sets (Fig 3D and S2 Table; Log2FC = 4.43, p = $7.06 \times 10^{-29}$) which was validated by qRT-PCR using two independent primer sets (Fig 3D). Unfortunately, we were unable to detect ZBTB16 protein in our cultures by western blotting as a suitable antibody is not available. Nonetheless, our integrated analysis of differential gene expression and differential

**Table 1. Cluster analysis of differentially bound CTCF binding peaks.**

| Chromosome | No. differential CTCF BS (1MB) | Cluster position start | Cluster position end | Annotated genes (^up, *down; p < 0.05) | Up (↑), Down (↓) |
|---|---|---|---|---|---|
| **Chr11** | 20 | 114067078 | 116512610 | ZBTB16^, CADM1* | 1↑, 19↓ |
| **Chr9** | 18 | 119808975 | 124907489 | ASTN1*, DBC1, CNTRL^, GSN, DAB2IP^, NDUFA8 | 3↑, 15↓ |
| **Chr17** | 17 | 46095759 | 51227805 | SKAP1, NGFR^, COL1A1, WFIKKN2, CA10* | 9↑, 8↓ |
| **Chr19** | 15 | 38481016 | 44278901 | SIPA1L3, FBXO17, CNTD2, SPTBN4, GRIK5, PSG5, XRCC1^, KCNN4^ | 9↑, 6↓ |
| **Chr17** | 13 | 77184053 | 81035156 | RBFOX3, CCDC40, SLC26A11, RPTOR, BAHCC1, CSNK1D, METRNL* | 11↑, 2↓ |
| **Chr8** | 13 | 37659173 | 40110928 | GPR124, LETM2^, HTRA4, TM2D2, ADAM5P | 8↑, 5↓ |
| **Chr10** | 12 | 84571415 | 87938467 | NRG3, GRID1 | 12↓ |
| **Chr18** | 12 | 5424957 | 6604578 | EPB41L3*, TMEM200C, L3MBTL4*, LOC100130480 | 12↓ |

The genomic location of differential CTCF binding peaks identified in ChIP-Seq datasets was identified using the bedtools cluster function (v2.26.0) with a maximum distance of 1 Mbp. The coordinates and number of differential CTCF binding peaks (up or down) within the cluster are indicated along with the genes located within the cluster (^up, *down; p < 0.05).

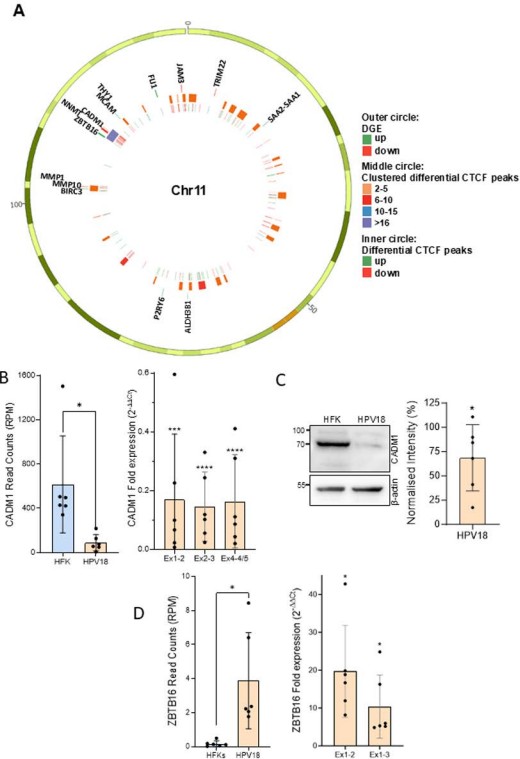

**Fig 3. HPV18-induced redistribution of a distinct CTCF binding cluster correlates with altered gene expression at the *CADM1* locus.** (**A**) Circos plot of Chromosome 11 (Chr11) showing differentially expressed genes (DGE; regularized log2 fold change cutoff (Log2FC) ≤ −1 or ≥1; outer circle), significantly different CTCF peaks obtained by ChIP-seq analysis (inner circle), and clusters of differential CTCF sites with a maximum distance of 1M base pairs (middle circle). (**B**) *CADM1* mRNA expression from RNA-Seq data (left panel) and qRT-PCR with primers designed to amplify exons (Ex) 1-2, exons 2-3 and exons 4-5 of *CADM1* (right panel). Data are the mean and standard deviation of six independent HFK donors. Statistical significance was calculated using a paired T-test *p < 0.05 (left) and a one sample T-test with theoretical mean of 1 *** p < 0.001, **** p < 0.0001 (right). (**C**) CADM1 protein expression was analysed by western blotting. A representative image is shown on the left. Protein levels were quantified using ImageJ and normalized to β-actin and expression in the corresponding HFK donor (untransfected). Data are the mean and standard deviation and statistical significance was calculated using a one sample T-test with theoretical mean of 100, p = 0.048. (**D**) *ZBTB16* mRNA expression from RNA-Seq data (left panel) and qRT-PCR with primers designed to amplify exons 1-2 and exons 1-3 (right panel). Data are the mean and standard deviation of six independent HFK donors. Statistical significance was calculated using a paired T-test, p < 0.02 (left) and a one sample T-test with theoretical mean of 1 *p < 0.05 (right).

CTCF binding peaks have identified a genomic locus on chromosome 11 which contains a cluster of CTCF binding sites that are significantly reduced in CTCF abundance which is coincident with profound locus-specific gene expression changes.

## HPV18 genome establishment induced epigenetic reprogramming of the *CADM1* gene locus

To elucidate the molecular basis of HPV18-induced *CADM1* transcriptional repression, we analysed chromatin structure by mapping histone post-translational modifications (PTMs) known to be associated with transcriptionally active (H3K4Me3 and H3K27Ac) and inactive (H3K27Me3) loci alongside a mark of transcription progression (H3K36Me3). Active promoters are generally enriched in H3K4Me3 while enhancers often have H3K4Me1/3 and H3K27Ac enrichment (reviewed by [33]). Analysis of these histone marks at the *CADM1*

promoter and upstream enhancer (annotated in www.enhanceratlas.org) demonstrated that enrichment of H3K4Me3 at the *CADM1* promoter, and H3K4Me3 and H3K27Ac marks within the upstream enhancer are notably reduced following HPV18 establishment (Fig 4). In contrast, the *CADM1* locus is devoid of repressive H3K27Me3 modifications in primary keratinocytes, the abundance of which is increased following HPV18 establishment, indicating specific epigenetic repression of *CADM1* expression (Fig 4). Interestingly, epigenetic alteration of the *CADM1* locus was coincident with epigenetic rearrangement of the neighbouring *ZBTB16* gene locus but in contrast to *CADM1*, the *ZBTB16* locus was increased in activating histone marks H3K4Me3 and H3K27Ac and decreased in H3K27Me3 following HPV18 establishment. These epigenetic alterations to the *CADM1/ZBTB16* loci are consistent with HPV18-induced local rearrangement resulting in repression of *CADM1* and activation of *ZBTB16* expression.

## HPV18 establishment induces topological rearrangement of the *CADM1* locus

The significant reduction of *CADM1* transcription and upregulation of *ZBTB16* following HPV18 establishment in all six independent primary keratinocyte cultures indicates a specific and targeted mechanism of HPV-induced host transcriptional reprogramming. This, combined with the loss of CTCF at the *CADM1* TSS and within the upstream enhancer elements, and the dramatic alteration of epigenetic marks within the *CADM1/ZBTB16* locus, led us to hypothesise that HPV18 establishment in primary keratinocytes induces CTCF-dependent topological rearrangement of this locus and contributes to the observed epigenetic reprogramming.

To test this hypothesis, we first analysed the CTCF binding peaks identified by ChIP-Seq to map the location, core sequence (presence of primary motif only or primary and secondary motif [34]) and orientation (sense or anti-sense) of the CTCF binding sites within the *CADM1* TSS and upstream enhancer element. We identified three CTCF binding sites at the *CADM1* TSS; sites 1 and 2 contain the primary CTCF binding motif only and were in the anti-sense orientation, facing towards the *CADM1* gene. Conversely, CTCF binding

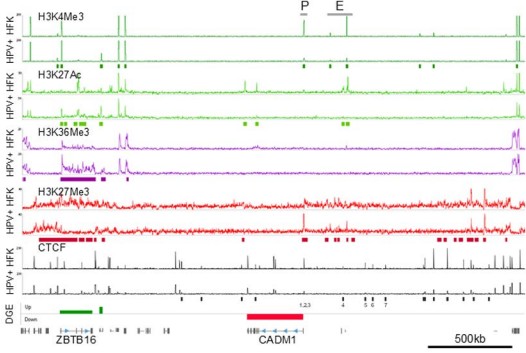

**Fig 4. HPV18 establishment induces epigenetic rearrangement of the *CADM1* locus.** ChIP-Seq analysis of histone modifications (H3K4Me3, dark green; H3K27Ac, light green; H3K36Me3, purple; H3K27Me3, red) and CTCF (black) in primary HFK donor 3 before (HFK) and after establishment of HPV18 episome replication (HPV+). Significantly differentially enriched peaks (p < 0.0001) were identified using MACS and indicated by the coloured blocks below the ChIP-Seq traces. Differential gene expression (DGE) from RNA-Seq analysis is shown in the lower track alongside annotation of *CADM1* and *ZBTB16* gene positions. The transcriptional promoter (P) and enhancer elements (E) of *CADM1* obtained from ENCODE are indicated. The epigenetic changes observed within the *CADM1/ZBTB16* locus were consistent in HFK donor 4.

site 3 at the *CADM1* TSS contains the primary and secondary motifs, indicating this site is higher affinity, and is orientated in the sense direction towards the upstream *CADM1* transcriptional enhancer (Fig 5A and 5B). None of the CTCF binding peaks identified at the *CADM1* TSS were found to be significantly altered in CTCF binding by HPV establishment in the two HFK donors tested (Fig 4). However, we also mapped four further CTCF binding sites, termed sites 4, 5, 6 and 7 situated upstream of the *CADM1* enhancer at −230 kb, −365 kb, −408 kb and −486 kb in relation to the *CADM1* TSS, respectively. All these CTCF binding sites contain the primary and secondary motifs suggesting they are high affinity binding sites that may contribute to the stabilisation of long-range chromatin interactions [35]. Notably, the upstream CTCF binding sites 4–7 were all significantly reduced in CTCF binding following HPV18 establishment (site 4, p = 0.0044; site 5, p = 0.002; site 6, p = 3.3e$^{-8}$; site 7, p = 2.97e$^{-8}$). CTCF binding sites 5 and 7 were found to be orientated in the sense

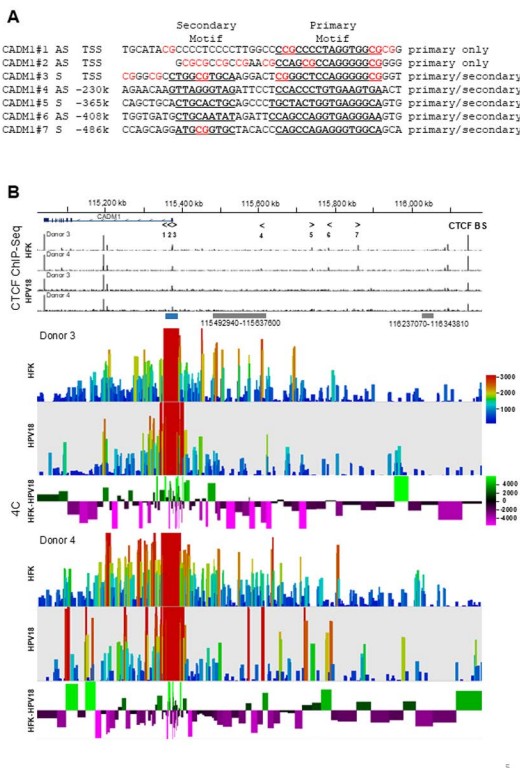

**Fig 5. HPV18 establishment induces topological rearrangement of the *CADM1* locus.** (**A**) CTCF bound regions at the CADM1 TSS and upstream transcriptional enhancer were identified from ChIP-Seq data and CTCF binding site identified based on similarity to previously determined binding motif [36]. Binding site number is indicated and corresponds to the binding site annotation in (**B**). Direction is shown as sense (S) or antisense (AS) alongside the nucleotide position (kb) in relation to the CADM1 TSS. Primary and secondary CTCF binding motifs are indicated in bold and underlined. CpG dinucleotides within the annotated binding sites are indicated in red. (**B**) Chromatin interactions with the CADM1 transcriptional start site at Chr11:115375845-115376296 (viewpoint indicted in blue) were analyzed by 4C in two independent HFK donors before and after HPV18 episome establishment. Annotated enhancer elements (grey boxes). The position and orientation of mapped reads was determined using the Probe Trend Plot feature of SeqMonk and shows relative read density over restriction fragments generated by *Dpn*II and *Bfa*I digest, correcting to the largest datastore. The geometric mean shows the average number of reads for each fragment and its immediate neighbouring fragments coloured according to a rainbow scale. Subtraction of normalised read counts of HFK-HPV18 minus HFK for each donor was calculated and shown below the 4C-Seq tracks for each donor. Green indicates a gain in interaction whereas purple indicates a loss.

direction, orientated away from the *CADM1* TSS, whereas sites 4 and 6 were orientated in the anti-sense direction, towards the *CADM1* TSS. We therefore predicted that CTCF-mediated chromatin looping could be facilitated between convergent CTCF sites 3 and either sites 4 or 6, which could bring the previously annotated distal enhancer element at position Chr11:115492940-115637600 ([www.enhanceratlas.org](http://www.enhanceratlas.org)) in close physical proximity to the *CADM1* TSS to stimulate transcription.

To determine whether cis- or trans- interactions exist between the *CADM1* TSS and any other genomic region in the cellular genome we used circular, chromatin conformation capture (4C) followed by next generation sequencing (4C-Seq). The experiment was designed to capture chromatin interactions with a viewpoint region containing the *CADM1* TSS (Chr11:115375845-115376296) created by restriction digest with *Dpn*II and *Bfa*I. Analysis of chromatin interactions in two independent primary keratinocyte donors revealed interactions between the *CADM1* TSS and the upstream enhancer element in both HFK donors tested (Fig 5B). Notably, in the presence of HPV18 genome replication, the majority of interactions between the *CADM1* TSS and the upstream enhancer element were greatly reduced. The reduction in interactions between the *CADM1* TSS and upstream enhancer regions is clearly visible in subtraction plots of 4C-seq reads from primary HFK minus HPV18 replicating samples (lower panels). The reduction in upstream enhancer interactions was observed in both donors tested, although the loss of interactions in donor 4 was less pronounced than in donor 3 and some adjacent areas showed increased interactions (Fig 5B). This is consistent with reorganisation of the 3D architecture of the whole upstream region. It is interesting to note that no HPV18-specific reads were detected in our 4C analysis, indicating that the HPV18 genome, maintained predominantly in an episomal state does not physically interact with the *CADM1* TSS.

## HPV18-induced loss of *CADM1* promoter-enhancer interactions precedes CpG hypermethylation of the *CADM1* locus

It has previously been shown that *CADM1* expression is significantly reduced in HPV-driven cancers [37,38] and this reduced expression correlates with hypermethylation of CpG islands at the *CADM1* TSS [39]. We therefore sought to determine whether the reduced *CADM1* expression and alterations to promoter-enhancer interactions were due to CpG hypermethylation at the *CADM1* TSS. This is particularly important since the CTCF binding sites present within the *CADM1* TSS (binding sites 1–3) each contain multiple CG dinucleotides, methylation of which would inhibit CTCF binding [40]. The enrichment of CpG methylation at the *CADM1* locus and surrounding genes was determined by adaptive Nanopore sequencing, which allows the direct detection of methylated cytosines at CpG sites without the need for bisulphite conversion [41]. Visualisation of CpG methylation within our region of interest (Chr11:112,500,000-117,500,000) in HFK and HPV18-HFK cultures (donors 3, 4 and 5) revealed a distinct area of predominantly unmethylated DNA at the *CADM1* locus, which was flanked by regions of higher CpG methylation up- and downstream (Fig 6A). The mean CpG methylation at the *CADM1* TSS in HFKs (donor 3, 11.34%; donor 4, 12.27%; donor 5, 20.41%) was not significantly altered following HPV18 establishment (donor 3, 26.76%; donor 4, 21.76%; donor 5, 15.25%; p = 0.078). While the mean percentage CpG methylation in each donor at the *CADM1* TSS was low, the abundance of CpG methylation detected at the neighbouring gene *APOA5* was significantly higher (donor 3, 75.4%; donor 4, 46.98%; donor 5, 76.23%; p < 0.0001, Fig 6B). These results suggest that the strong repression of *CADM1* expression by HPV18 genome replication in primary HFKs is not due to increased CpG methylation of the *CADM1* TSS, as previously demonstrated in HPV-driven cancers.

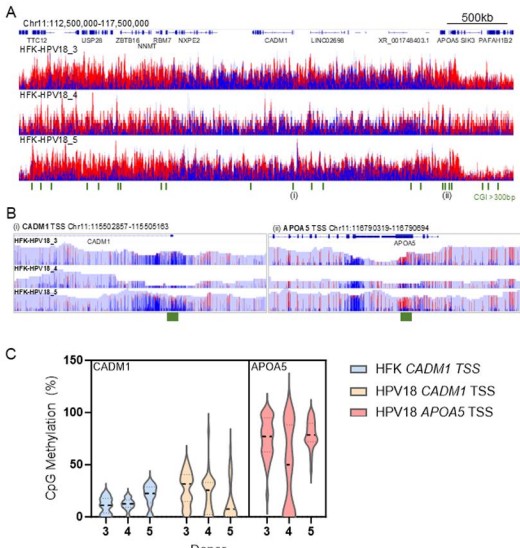

**Fig 6. Low abundance of CpG methylation at the *CADM1* locus following establishment of HPV18 replication.** Methylation at CpG dinucleotides was determined by adaptive Nanopore sequencing in donors 3, 4 and 5. (**A**) Histogram of aligned reads (depth 0–25 counts) is shown, CpG dinucleotides are coloured by 5mC modified CpG (blue indicates unmethylated and red indicates methylated cytosines). The position of genes within the region shown are indicated at the top. CpG islands of >300 bp were identified using Encode and indicated by the green bars below the figure. (**B**) Regions of interest at the *CADM1* TSS (i) and *APOA5* TSS (ii), as depicted on (A) are shown. (**C**) Violin plots showing the distribution of % CpG methylation at CpG dinucleotides at the *CADM1* TSS in HFK donors (blue) and HPV18-HFK (Orange) alongside % CpG methylation at the upstream *APOA5* TSS in HPV18-HFK (red). Data median are indicated by the bold dotted line and quartiles by fine dotted lines. Any difference in % CpG methylation at the *CADM1* TSS between HFK and HPV18-HFK is not significant (p = 0.078), whereas % CpG methylation is significantly higher at the *APOA5* TSS in comparison to the *CADM1* TSS in HPV18-HFK (p < 0.0001, two-way ANOVA with multiple comparisons).

## Discussion

The establishment of persistent oncogenic HPV replication and *in vitro* immortalisation of primary keratinocytes is facilitated by manipulation of the host transcriptome to alter the immune response to infection and drive a rebalancing of cellular growth and differentiation. Studies of HPV-induced transcription manipulation of the host in physiological models of oncogenic HPV replication have defined key events in HPV establishment and host cell immortalisation [3,4,6,42]. Comparison of differential gene expression following HPV16-mediated *in vitro* immortalisation of primary cervical keratinocytes has previously revealed a strong concordance with aberrant gene expression in HPV-driven cervical cancers [3]. It has also been shown that HPV positive tumours have a distinct pattern of differential gene expression, mostly driven by E6/E7 expression, compared to HPV- tumours at the same anatomical site and a consistent pattern of aberrant gene expression is noted in HPV+ tumours from distinct anatomical sites [43]. These findings provide strong evidence that HPV-driven host transcription reprogramming is important for persistent HPV replication as well as initiation of tumourigenesis and subsequent maintenance of tumour growth.

In this study we use a panel of six genetically distinct primary HFKs to define gene expression changes that occur following establishment of HPV18 episome maintenance replication. RNA-Seq analysis revealed a robust and consistent pattern of differential gene expression across HFK donors. Comparison of the gene expression changes that occur in all six HFK donors reveals a discrete subset of significantly altered genes; a total of 628 genes were

upregulated and 348 genes downregulated. A lower number (228) of genes were differentially expressed upon establishment of HPV18 episomes in HFKs and analysed using a microarray assay [44]. Of these, 44 matched the differentially expressed genes in our study, with all 44 being regulated in the same direction. In an HPV16 infection model, based on infection of primary foreskin keratinocytes with HPV16 virions, an immediate early event in the establishment of HPV16 replication is E7 mediated deregulation of pocket protein-controlled pathways [45]. Significant deregulation of these pathways is also maintained in our HPV18-HFK models, indicating a requirement for maintenance of long-term growth of cells harbouring oncogenic HPV. That CADM1 deregulation occurs in our models and in cancers strongly suggests that this is a virus-induced perturbation necessary for longer term and/or persistent infection. Interestingly this study also identified a similar number of differentially expressed genes to those identified in our study following HPV16-mediated HFK immortalisation, when the same false discovery rate (FDR) and statistical thresholds are applied, with some overlap with the differentially expressed genes identified in our study (e.g., *UHCL1*, *RNF212*, and *NEFH*) [45].

We hypothesised that the mechanisms driving HPV-manipulation of host gene expression in our model of productive infection are likely to be different to the less dynamic mechanisms of sustained aberrant gene expression observed in HPV-driven cancers. GSEA highlighted pathways targeted by HPV18 consistent with previous reports, including E2F signalling, $G_2$/M checkpoint response and various immune signalling pathways [44,46]. Notably, some of the gene expression changes we observed in our pre-disease model of HPV replication are also known to occur with high frequency in HPV-driven cancers. For example, *CADM1* expression was reduced in all six HFK donors following HPV18 replication establishment. This gene has been shown to be strongly downregulated in HPV+ cancers [37–39]. We now show that HPV18-induced repression of *CADM1* transcription is an early event in HPV-driven pathogenesis and occurs prior to virus-induced cell transformation.

We and others have previously shown that oncogenic HPV establishment in primary keratinocytes results in a post-transcriptional increase in CTCF protein abundance [18,31], although further analysis of mulitple HFK donors in this study revealed no difference in CTCF transcript or protein abundance after HPV18 establishment. Nonetheless, CTCF plays a key role in the regulation of HPV and host transcription through the regulation of epigenetic chromatin status and stabilisation of long-range chromatin interactions, often between promoter and enhancer elements to drive transcription [18,19,47]. To determine whether HPV manipulation of CTCF function is in part responsible for HPV18-induced differential host gene expression, we performed ChIP-Seq analysis to map CTCF binding sites and key epigenetic marks in the host genome following HPV18 establishment in two independent HFK donors. While the overall number of CTCF binding peaks did not change, the distribution of a sub-set of CTCF bound peaks was significantly altered; 900 distinct CTCF peaks displayed a significant gain (n = 504) or loss (n = 396) of CTCF binding with an apparent bias in loss of binding within gene desert/intergenic regions. Integration of the differentially bound CTCF peaks with differential gene expression data identified gene loci that contained clusters of differentially bound CTCF peaks coincident with significant changes to gene expression. Notably, the *CADM1/ZBTB16* gene locus contained 20 differentially bound CTCF sites predominantly located up-stream of the *CADM1* gene body including within the upstream transcriptional enhancer of *CADM1*. General loss of CTCF binding in this region in HFKs with replicating HPV18 genomes was coincident with increased *ZBTB16* expression and decreased *CADM1* expression in all six HFK donors tested. The loss of CTCF binding in this region was also coincident with altered epigenetic status within the *CADM1* TSS and upstream enhancer from an active to repressed chromatin state. In the two donors analysed,

HPV18 replication caused a loss of active chromatin marks H3K4Me3 and H3K27Ac and gain of repressive H3K27Me3 within the *CADM1* TSS and upstream enhancer. Whether this alteration of epigenetic marks is a cause or consequence of CTCF binding disruption at this locus is not yet clear. However, CTCF binding is important for epigenetic insulation between active and inactive gene regions [48]. It is therefore possible that initial disruption of CTCF binding upstream of the *CADM1* gene removes epigenetic insulation from the neighbouring repressed gene region thus allowing the spread of repressive H3K27Me3 marks into the *CADM1* locus causing a downregulation of *CADM1* expression, a hypothesis we are currently testing.

Binding of CTCF to target sites can be disrupted by methylation of CpG dinucleotides within the binding footprint [49]. Notably, it has also been shown that downregulation of *CADM1* in HPV-driven cancer is due to hypermethylation of a CpG island at the *CADM1* TSS [37,38,50] and that sequential hypermethylation at the *CADM1* TSS correlates with HPV-driven cancer stage [12,13,39]. To determine whether HPV18-induced disruption of CTCF binding at the *CADM1* locus could be due to increased CpG methylation of these binding sites, particularly within the upstream transcriptional enhancer, we mapped binding sites by comparison of the genomic sequences within the binding footprints identified in our ChIP-Seq data with the defined CTCF binding motif (Jasper.com). While the CTCF binding sites at the *CADM1* TSS each contained multiple CpG dinucleotides and therefore CTCF binding at these sites could be sensitive to increased CpG methylation, the upstream sites (4, 5 and 6) did not contain CpG dinucleotides. HPV18-induced loss of CTCF binding at sites 4, 5 and 6 in the *CADM1* enhancer is therefore not predicated in alterations in CpG methylation. To determine whether loss of CTCF binding at sites 1, 2 and 3, situated at the *CADM1* TSS was due to HPV18-induced hypermethylation of the *CADM1* TSS, we analysed CpG methylation levels by Oxford Nanopore sequencing. These experiments revealed a distinct area of hypomethylated DNA at the *CADM1* locus that was flanked by hypermethylated regions in HPV18-genome containing HFKs harvested from three individual donors. Furthermore, the difference in CpG methylation in the three donor HFK cultures analysed before and after HPV18 establishment was not significantly different. These results are consistent with a previous study which assessed the CpG methylation status within 14 key genes, including *CADM1*, in a model of E6/E7-driven primary keratinocyte immortalisation. Only a small increase in CpG methylation of the *CADM1* promoter between HPV negative controls and cells harvested between 14–80 passages after E6/E7-mediated immortalisation was observed; indeed increased methylation was only observed in the control HPV+ cancer cell line (SiHa) [13]. We therefore conclude that the rearrangement of CTCF binding and altered epigenetic regulation of the *CADM1* locus is a pre-cursor to hypermethylation of the *CADM1* promoter frequently observed in HPV-driven cancers. Interestingly, a high proportion of CpG islands contain TSSs and it is generally accepted that unmethylated CpG islands act as a nucleation site for the recruitment of histone modifying enzymes such as KDM2A which facilitates enrichment of histone H3 lysine 4 tri-methylation (H3K4Me3) thereby maintaining an active chromatin state [51,52]. The mechanisms of *de novo* methylation of CpG islands are not fully understood but a strong correlation with loss of H3K4Me3 and a gain of polycomb repressive complex (PRC)-catalysed histone modifications including histone H3 lysine 27 trimethylation (H3K27Me3), creating a repressed chromatin state has been observed [52]. Evidence suggests that this PRC-mediated H3K27Me3 enrichment is a pre-requisite for *de novo* CpG island methylation in embryonic development and in cancer [53] and creates a constitutively silent chromatin state.

In summary, we have shown that replication of HPV18 in primary HFKs isolated from six independent donors causes a consistent pattern of differential gene expression in the host cell. Integration of RNA-Seq data with CTCF ChIP-Seq data revealed that many of the significantly

altered genes map to genomic loci that contain discrete clusters of differentially bound CTCF sites. The *CADM1* locus on chromosome 11 contains 20 differentially bound CTCF binding sites, 19 of which display a significant reduction of CTCF binding in cells with persistently replicating HPV18 genomes. This loss of CTCF binding is coincident with a switch in epigenetic mark enrichment, from an active to inactive chromatin state, and with reduced *CADM1* expression, but not with increased CpG hypermethylation as has been described in HPV-driven cancers. We have demonstrated that rearrangement of CTCF binding induced a structural change upstream of the *CADM1* genebody, such that physical contacts between the transcriptional enhancer and *CADM1* TSS are disrupted. Although other studies have shown that *CADM1* expression negatively correlates with HPV-driven cancer stage, we now provide evidence that *CADM1* gene expression is targeted by oncogenic HPV prior to cellular transformation, suggesting that *CADM1* repression is an important early event in HPV establishment. Functionally, *CADM1* is a member of the immunoglobulin superfamily of transmembrane glycoproteins. The extracellular domain mediates interactions of the keratinocyte with extracellular matrix and neighbouring cells including dendritic cells, NK and CD8+ T cells, while the intracellular domain contains protein 4.1-binding and PDZ-binding motifs which, upon binding intracellular partners, regulates cell motility. HPV-mediated repression of *CADM1* is therefore likely to be important in persistence of infection by altering both epithelial tissue integrity and immune cell attraction and activation within the lesion.

## Materials and methods

### Ethics statement

Normal primary foreskin keratinocytes (HFKs) were harvested from neonatal foreskin epithelia, collected anonymously with formal written parental/guardian consent (HSE ethical approval number 06/Q1702/45; reviewed by South Central – Hampshire A Research Ethics Committee, UK).

### Primary keratinocyte culture and transfection

The transfection of HFKs with recircularized HPV18 genomes was performed in S. Roberts' laboratory as previously described [19,54]. To eliminate donor-specific effects, primary cells from six foreskin donors were used: five isolated in house and one commercially available (Lonza).

### RNA-sequencing

Primary HFKs and HPV18-transfected isogenic lines ($1 \times 10^5$) were cultured on γ-irradiated J2 fibroblasts in epidermal growth factor containing E medium [25] in 10 cm dishes until 80% confluent. J2 fibroblasts were then removed and HFKs harvested by trypsinisation. Cells were pelleted, washed with PBS, snap frozen and stored at –80 °C. RNA was extracted using the RNeasy Mini Kit following the manufacturer's protocol (Qiagen) and DNase I treated. Libraries were prepared using TruSeq Stranded mRNA Library Prep kit for NeoPrep (Illumina) using 100 ng total RNA. Libraries were pooled and run as 75 cycle pair end reads on a NextSeq 550 (Illumina) using a high output flow cell.

Sequencing reads were aligned to GRCh37 human genome merged with HPV18 genome (AY262282.1) using STAR aligner (v2.5.2b) [55]. Reads mapping to genes were counted by the same software. Normalisation of read counts and differential expression analysis between HFK and HPV18 samples was performed with DESeq2 (v.1.26.0) R Bioconductor package [56], taking into account the paired sample design of the experiment. Gene set enrichment analysis was done using GAGE (v.2.36.0) R Bioconductor package [57] with MSigDB gene

set database. The computations were performed on the CaStLeS infrastructure (http://doi.org/10.5281/zenodo.3250616) at the University of Birmingham.

The PlotPCA function in DESeq2 was used to calculate the principal components of rlog transformed RNA-Seq expression data of the 500 most variable genes. The ClusterSignificance [58] R Bioconductor package was used to evaluate statistical significance of group separation compared to permuted data. The P-value was calculated using $10^4$ permutations.

## Chromatin immunoprecipitation (ChIP) and next generation sequencing

$1–2 \times 10^7$ cells were fixed in 1% formaldehyde for 3 mins at room temperature, quenched in 0.25 M glycine and washed in ice cold PBS. Chromatin was immunoprecipitated with 2–10 μg of antibody specific for CTCF (Cell Signaling #D31H2-XP) and histone modifications H3K4Me3 (Upstate: #04-745), H3K27Ac (Abcam: ab4729), H3K36Me3 (Cell Signaling: 4909S) and H3K27Me3 (Upstate: #07-449) as previously described [59]. ChIP and respective input samples were used for generation of ChIP-Seq libraries as described [60]. Briefly, 2–10 ng DNA was used in conjunction with the NEXTflex Illumina ChIP-Seq library prep kit (Cat# 5143-02) as per the manufacturer's protocol. Samples were sequenced on a HiSeq 2500 system (Illumina) using single read (1 x 50) flow cells. Sequencing data was aligned to the GRCh37 human genome using Bowtie [61] with standard settings and the -m1 option set to exclude multi mapping reads [59].

## Differential ChIP-seq analysis

CTCF enriched sites (CTCF peaks) on the human genome were called in all samples using MACS1.4 [62]. Peaks were first filtered to be present in at least two samples and were subsequently merged into a stringent CTCF peak set using the Bioconductor R package DiffBind (v3.6.5) [63]. Differential sites from CTCF ChIP replicates were detected and annotated with diffReps [64] using presets of negative binomial statistics and nucleosome size detection mode parameters. Differential sites were then reported if overlapping with the stringent peak set to exclude background detection. Normalized heatmap visualization of differential sites (log2 cutoff −1 or 1) was generated with EaSeq (v1.1.1) [65]. Differential CTCF peaks were then clustered using the bedtools cluster function (v2.26.0) [66] with a maximum distance of 1 Mbp.

Differential ChIP-seq analysis of post-translational histone modifications was performed similarly to CTCF using diffReps [64] with negative binomial detection presets for replicates. Differential sites for activating histone modifications (H3K4Me3 and H3K27Ac) were detected with "peak" mode presets while broader differential regions of the repressive mark H3K27Me3 and the transcription-associated mark H3K36Me3 were detected with "block" mode presets.

## Integration of RNA-seq and ChIP-seq data

Differentially expressed genes and differential CTCF binding site clusters as well as sites of differential histone modifications were associated using the bedtools closest function (v2.26.0) [66]. Circos plot visualization of differentially expressed genes, individual differential CTCF binding sites and CTCF clusters was generated with Circos (v0.69-6) [67].

## Quantitative reverse transcriptase PCR (qRT-PCR)

DNase I treated RNA (3 μg) was used for random primed cDNA synthesis using Superscript III (Invitrogen) according to the manufacturer's instructions. qPCR was performed with 40 cycles of 95 °C, 30 s; 60 °C, 60 s; 72 °C, 40 s followed by a thermal dissociation curve for QC analysis using a Stratagene Mx3005P detection system with SyBr Green incorporation and the primers listed in Table 2.

**Table 2. qRT-PCR primers.**

| Gene | Forward 5'-3' | Reverse 5'-3' |
|---|---|---|
| CADM1 (exon 1-2) | ATGGCGAGTGTAGTGCTGC | GATCACTGTCACGTCTTTCGT |
| CADM1 (exon 2-3) | GACGTGACAGTGATCGAGGG | GGGATCGGTATAGAGCTGGCA |
| CADM1 (exon 4-4/5) | GTCCCACCACGTAATCTGATG | CCACCTCCGATTTGCCTTTTA |
| ZBTB16 (exon 1-2) | TGTGGGGTCGAGCTTCCTGA | GCACCCGTACGTCTTCATCCC |
| ZBTB16 (exon 1-3) | TGTGGAGCAGCACAGGAAGC | CCTTCGAAAACTGTGCACCGC |
| β-actin | GCTGTGCTATCCCTGTACGC | CAGGAAGGAAGGCTGGAAGA |

## Western blotting and antibodies

Cells were lysed in urea lysis buffer (8 M Urea, 100 mM Tris-HCl, pH 7.4, 14 mM β-mercaptoethanol, protease inhibitors) and protein concentration determined by Bradford assay. Equal amounts of protein were separated by SDS-PAGE and western blotting carried out using conventional methods. CADM1 protein was detected using SynCAM Rabbit polyclonal antibody (Abcam #ab3910; 1:1000), CTCF detected with Active Motif (#61311; 1:1000), GAPDH with Santa Cruz antibody (#0411; 1:5000) and β-actin detected with Sigma antibody (#A5441; 1:5000).

## Circularised chromosome conformation capture (4C-seq)

4C-Seq was carried out in three independent primary HFK donors and isogenic HPV18 transfected lines using a previously described protocol [68,69]. CADM1 promoter interacting fragments were captured using a 451 bp *Dpn*II fragment containing the CADM1 promoter, prior to digestion with *Bfa*I. Cells were passed through a 70 μm filter, counted and $1 \times 10^7$ cells were fixed in 2% formaldehyde/10% FCS in PBS for 10 mins at RT before quenching with 0.125 M glycine on ice. Cells were pelleted by centrifugation at $400 \times g$ for 8 mins at 4 °C and resuspended in 5 mL ice cold lysis buffer (50 mM Tris-HCl, pH 7.5, 150 mM NaCl, 5 mM EDTA, 0.5% NP-40, 1% Triton-X-100, 1x protease inhibitor cocktail (Roche)) and incubated on ice for 10 mins. Nuclei were pelleted at $750 \times g$ for 5 mins at 4 °C and snap frozen before storage at –80 °C.

**Primary digestion and ligation.** Pellets were resuspended in 500 μL 1x *Dpn*II restriction enzyme buffer (NEB) containing 0.3% SDS and incubated at 37 °C with shaking at 900 rpm. Triton-X-100 was added to a final concentration of 0.625% and the samples incubated for a further 1 hr at 37 °C with shaking at 900 rpm. 200 U of *Dpn*II was then added and samples incubated at 37 °C for 4 hrs with shaking at 900 rpm. A further 200 U of *Dpn*II was added and samples digested overnight at 37 °C with shaking and a final 200 U *Dpn*II was then added for a final 4 hrs to ensure complete digestion. Following restriction enzyme inactivation, digestion efficiency was determined by agarose gel electrophoresis following Proteinase K digestion of undigested and digested aliquots. Proximity ligation was carried out at 16 °C overnight by the addition of 1x ligase buffer to 7 mL and 10 μL T4 DNA Ligase (NEB) and ligation efficiency determined by agarose gel electrophoresis following Proteinase K digestion of an aliquot.

Proteinase K was added to a final concentration of 40 μg/mL and samples incubated overnight at 65 °C to reverse crosslinks. 40 μg/mL RNase A was added and samples incubated at 37 °C for 45 mins before extraction with phenol-chloroform-isoamylalcohol (24:24:1) and ethanol precipitation. Pellets were resuspended in 150 μL 10 mM Tris-HCl, pH 7.5.

**Secondary digestion and ligation.** To 150 μL of each sample, 50 μL 10x restriction enzyme buffer, 50 U of *Bfa*I enzyme and 250 μL ddH$_2$0 were added and samples incubated overnight at 37 °C. Digestion efficiency was determined as previously described and restriction enzyme heat inactivated before samples were made up to 14 mL with 1x ligase buffer and 20 μL ligase added. Ligation was carried out at 16 °C overnight. DNA was ethanol precipitated and

samples purified using a QIAquick PCR purification kit (Qiagen) and eluted in 150 μL Tris-HCl, pH 7.5.

**PCR amplification of 4C template.** Captured fragments were amplified by inverse PCR using primers that amplify outwards from the bait region (Table 3). Forward primers included a 5' overhang containing the Illumina P5 sequence adapter and a unique 'barcode' sequence and encompassed the primary *Dpn*II restriction site within the *CADM1* promoter bait. The common reverse primer included a 5' overhang containing the Illumina P7 sequence adapter and were designed to bind less than 100 bps from the secondary *Bfa*I restriction site in the *CADM1* promoter bait. PCR was carried out using 11.2 μL Expand Long Template Polymerase (Roche) with 3.2 μg template, 1.12 nmol of P5 and P7 primers, 0.2 mM dNTPs in an 800 μL reaction divided into 16 PCR tubes for 2 min 94 °C, 29 cycles of 10 s 94 °C, 1 min 55 °C and 3 mins 68 °C, followed by 5 mins at 68 °C. The reactions were pooled and purified using a High Pure PCR Product Purification Kit (Roche).

**Sequencing, demultiplexing and mapping.** The samples underwent 101 base pair, single-end sequencing on a HiSeq 2500, to generate FASTQ files using standard Illumina base-calling pipeline software. The samples were combined in one sequencing lane and so required demultiplexing post-sequencing. The library construction positioned the barcodes "in-line" with the target sequences and comprised the first 6 bases of each read. A custom Perl script was used to perform the demultiplexing, which identified FASTQ reads containing expected barcode sequences. The custom script performed additional validation by selecting only reads that conformed to the library design, by selecting for reads that contained a fixed sequence (TGAGCATACCCTCCTCGATC) immediately adjacent to the barcode. Furthermore, sequences which read-through immediately into the bait sequence (TTCTAGAGGGGAAGAAAATAAGTA) were discarded.

The FASTQ files were evaluated for sequence quality using FastQC (https://www.bioinformatics.babraham.ac.uk/projects/fastqc/). We confirmed that the reads contained human DNA and were free from likely sources of genetic contamination using FastQ Screen v0.14.0 [70]. Prior to mapping, the FASTQ files were quality trimmed with Trim Galore v0.6.2 (https://www.bioinformatics.babraham.ac.uk/projects/trim_galore/), using Cutadapt v1.18 [71]. This step removed low-quality and adapter sequences. The resulting FASTQ files were mapped against human genome assembly GRCh37 and the AY262282.1 genome assembly of HPV18 using Bowtie2 (v2.3.2) with default mapping parameters.

**Validating read position and orientation.** In accordance with our experimental protocol, canonical 4C ligations should generate reads derived from genomic regions adjacent to

**Table 3. 4C primers.**

| Primer ID | Barcode | Sequence (5'-3') |
|---|---|---|
| **CADM1 P5 TSBC02** | **CGATGT** | AATGATACGGCGACCACCGAGAACACTCTTTCCCTACAC-GACGCTCTTCCGATCT*CGATGT***TGAGCATACCCTCCTCGATC** |
| **CADM1 P5 TSBC04** | **TGACCA** | AATGATACGGCGACCACCGAGAACACTCTTTCCCTACAC-GACGCTCTTCCGATCT*TGACCA***TGAGCATACCCTCCTCGATC** |
| **CADM1 P5 TSBC05** | **ACAGTG** | AATGATACGGCGACCACCGAGAACACTCTTTCCCTACAC-GACGCTCTTCCGATCT*ACAGTG***TGAGCATACCCTCCTCGATC** |
| **CADM1 P5 TSBC06** | **GCCAAT** | AATGATACGGCGACCACCGAGAACACTCTTTCCCTACAC-GACGCTCTTCCGATCT*GCCAAT***TGAGCATACCCTCCTCGATC** |
| **CADM1 P7** | | CAAGCAGAAGACGGCATACGA**CGAAATTCTCTTTGCTTTCT** |

4C libraries generated from each sample (2 independent HFK donors; HPV18- and HPV18+) with were amplified with sense a primer containing an Illumina P5 adapter (adapter) and unique TSBC barcode (italics) and a common primer containing a P7 Illumina adapter (underlined). *CADM1*-specific viewpoint priming sites are indicated in bold.

*Dpn*II cut sites. Moreover, forward reads should map preferentially to the 5' end of *Dpn*II fragments, while reads in the reverse orientation should map to the 3' ends. In contrast, reads should not exhibit such positional and orientation biases with respect to *Bfa*I cut sites. To confirm these expectations, *in silico Dpn*II/*Bfa*I genome digestions were performed on human genome GRCh37 FASTA files with the Digester script from HiCUP v0.7.2 [72]. The position and orientation of the mapped reads were determined using the genome browser SeqMonk (https://www.bioinformatics.babraham.ac.uk/projects/seqmonk/). After importing the BAM files (generated by mapping FASTQ files with Bowtie 2) and the HiCUP-generated genome digest file into SeqMonk, the "Probe Trend Plot" feature of SeqMonk calculated the relative read density over every restriction fragment in the genome. SeqMonk was also used to quantitate the number of mapped reads aligning to each restriction fragment (or contiguous merged restriction fragments).

**4C subtractive quantitation analysis.** Read quantitation comparisons were performed using the bioinformatics tool SeqMonk. Firstly, the SeqMonk pipeline "Even Coverage Probe Generator" was run, using default parameters, to create tiled windows across the Human GrCh37 reference genome. These windows varied in length, so that each one contained the same number of reads (when pooling reads across all the datasets). As expected, this pipeline produced smaller windows proximal to the 4C bait, with these window lengths increasing steadily with distance from the bait. The number of reads per window was then re-quantitated for each dataset and, to enable direct comparisons between datasets, the read counts were normalised by library size. This normalisation step effectively sets the total number of reads to be the same for each dataset. SeqMonk was then used to subtract the normalised read count for every window in the pre-infection control dataset from its corresponding window in the relevant post-infection sample dataset. This paired control/sample subtraction step was repeated for every patient. The results were displayed by SeqMonk as a bar chart adjacent to the associated genome annotation.

## Adaptive Nanopore sequencing and CpG methylation analysis

Sequencing was carried out on genomic DNA extracted from HFK-HPV18 donors 3, 4 and 5 using a Qiagen Blood DNA extraction kit. Approximately 1 μg DNA was prepared using the Oxford Nanopore LSK109 library preparation protocol according to manufacturer's instructions. 150 fmol of completed library was sequenced on Nanopore Minion flowcells using R9.4.1 chemistry using an adaptive sampling protocol. The contigs of the GRCh38 reference genome and HPV18 genomes were merged and used to provide a custom genome for the adaptive sampling. Custom BED files targeting the regions of interest (ROI, S1 and S2 Datas) was designed according to adaptive sampling parameters and used to control the sequencing run. The adaptive sampling run was set to "enrich" and the flowcells run for 72 hours, with supplemental flush and reload if necessary. Data was basecalled using Dorado 0.6 using the R9.4.1 methylation specific model (dna_r9.4.1_e8_hac@v3.3) with 5 mC and 5 hmC calling switched on. This was then aligned to the custom GRCh38+HPV16/18 reference genome using minimap 2.12 (parameters: -ax map-ont), sorted and indexed with Samtools 1.16. BAM files were inspected manually with IGV v2.16 with methylation tagging visibility switched on.

## Supporting information

**S1 Table. RNA-sequencing analysis.** Table shows total number of reads per sample alongside number of uniquely mapped reads, reads mapped to the HPV18 genome (AY262282.1) and the number of HPV-host fusion transcripts identified.
(DOCX)

**S2 Table. Genes upregulated in HFK-HPV18 compared to HFK.** Mean sequencing counts in HFK and HFK-HPV18 are shown alongside log2 FC in gene expression and adjusted p value for each gene identified.
(XLSX)

**S3 Table. Genes downregulated in HFK-HPV18 compared to HFK.** Mean sequencing counts in HFK and HFK-HPV18 are shown alongside log2 FC in gene expression and adjusted p value for each gene identified.
(XLSX)

**S4 Table. ChIP-Seq analysis of CTCF bound chromatin regions.** Total CTCF binding peaks identified is given in the first tab. The second tab shows peaks that were differentially bound between HFK and HFK-HPV18 (mean of two donors). Data show the chromosomal location of identified peaks alongside the peak height, log2FC (up or down), neighbouring gene and genomic feature.
(XLSX)

**S5 Table. Integration of differential gene expression and differential CTCF peak annotation.** RNA-Seq data (blue) and ChIP-Seq data (green) were integrated to determine the distance of genes (red) that were altered at least 2-fold (up and down) from the nearest at least 2-fold differentially bound CTCF peak.
(XLSX)

**S6 Table. Cluster analysis of CTCF binding sites.** The bedtools cluster function was used to identify all loci with multiple differentially bound CTCF peaks (Table 1), integrated with corresponding differential expression (Table 2).
(XLSX)

**S1 Fig. HPV18 DNA is maintained as extrachromosomal episomes in HFK-HPV18 cultures.** (**A**) Southern blot analysis of DNA extracted from HFK-HPV18 donors 3, 4, 5 and 6. Samples digested with *Hin*dIII show open circle (OC) and supercoiled (SC) forms, whereas digestion with *Eco*RI linearises the HPV18 episomes as linear (L) forms. Southern blot analysis of donors 1 and 2 was previously described [48,49]. (**B**) Alignment of HPV18 transcripts identified by RNA-Seq to the linearized HPV18 genome (annotated below) for HFK-HPV18 donors 1-6. Image created using Integrated Genome Viewer.
(TIF)

**S2 Fig. Differential gene expression analysis following HPV18 replication establishment.** Volcano plot of gene expression profiles obtained from differential gene expression analysis of donor matched HFK in comparison to HFK-HPV18 (donors 1-6). Thresholds of Log2FC = 1 and adjusted p-value = 0.05 are indicated by the dotted lines. Significantly altered genes are indicated by red dots.
(TIF)

**S3 Fig. HPV18 genome replication results in significantly reduced CADM1 gene expression.** Alignment of RNA-Seq data to the CADM1 open reading frame (exons indicated below the image) in HFK donors 1-6 before and after HPV18 genome replication establishment. Paired tracks for each donor are group scaled as indicated on the right. Dotted line indicates truncation of the tracks within the large intron between exon 1-2. Image created using Integrated Genome Viewer.
(TIF)

**S4 Fig. HPV18-induced redistribution of CTCF binding clusters correlates with altered gene expression.** Data show individual read counts (reads per million; RPM) of significantly

altered genes and chromosomal location in six HFK donors following HPV18 establishment located within clustered differentially bound CTCF binding sites (Table 1). The bars indicate the mean gene expression and error bars indicate standard deviation. Statistical significance was calculated using a paired T-test *p < 0.05 **p < 0.01, ****p < 0.0001.
(TIF)

**S1 Data.**
(XLSX)

**S2 Data.**
(TXT)

**S3 Data.**
(TXT)

## Author contributions

**Conceptualization:** C. David Wood, Michelle J. West, Andrew Beggs, Adam Grundhoff, Boris Noyvert, Sally Roberts, Joanna L. Parish.

**Data curation:** Karen Campos-León, Jack Ferguson, Thomas Günther, Steven W. Wingett, Christy S. Varghese, Leanne S. Jones, Joanne D. Stockton, Andrew Beggs, Boris Noyvert, Joanna L. Parish.

**Formal analysis:** Karen Campos-León, Jack Ferguson, Thomas Günther, Steven W. Wingett, Selin Pekel, Christy S. Varghese, Csilla Várnai, Andrew Beggs, Boris Noyvert, Joanna L. Parish.

**Funding acquisition:** Sally Roberts, Joanna L. Parish.

**Investigation:** Adam Grundhoff, Sally Roberts, Joanna L. Parish.

**Methodology:** Karen Campos-León, Thomas Günther, C. David Wood, Steven W. Wingett, Selin Pekel, Christy S. Varghese, Leanne S. Jones, Joanne D. Stockton, Csilla Várnai, Michelle J. West, Andrew Beggs, Adam Grundhoff, Boris Noyvert, Joanna L. Parish.

**Project administration:** Sally Roberts, Joanna L. Parish.

**Resources:** Michelle J. West, Adam Grundhoff, Joanna L. Parish.

**Supervision:** Karen Campos-León, Jack Ferguson, C. David Wood, Csilla Várnai, Michelle J. West, Adam Grundhoff, Boris Noyvert, Sally Roberts, Joanna L. Parish.

**Validation:** Karen Campos-León, Jack Ferguson, Leanne S. Jones, Boris Noyvert.

**Visualization:** Thomas Günther, Steven W. Wingett, Selin Pekel, Csilla Várnai.

**Writing – original draft:** Sally Roberts, Joanna L. Parish.

**Writing – review & editing:** Karen Campos-León, Jack Ferguson, Thomas Günther, Boris Noyvert, Joanna L. Parish.

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
