## [Decision Letter · Decision Letter 0]

18 Sep 2024

Dear Dr. Parish,

Thank you very much for submitting your manuscript "Repression of CADM1 transcription by HPV type 18 is mediated by three-dimensional rearrangement of promoter-enhancer interactions" for consideration at PLOS Pathogens. As with all papers reviewed by the journal, your manuscript was reviewed by members of the editorial board and by several independent reviewers. The reviewers appreciated the attention to an important and exciting topic but identified some aspects of the manuscript that should be improved. Based on the reviews, we are likely to accept this manuscript for publication, providing that you modify the manuscript according to the review recommendations.

Sincerely,

Cary A. Moody

Academic Editor

PLOS Pathogens

Patrick Hearing

Section Editor

PLOS Pathogens

Michael Malim

Editor-in-Chief

PLOS Pathogens

orcid.org/0000-0002-7699-2064

Reviewer Comments (if any, and for reference):

Reviewer's Responses to Questions

**Part I - Summary**

Reviewer #1: Campos-Leon et al. describe the effects of CTCF, whose levels are increased in HPV 18 positive cells, on cellular gene expression. Previous studies from this group examined the effects of CTCF on HPV gene expression through interactions with YY1 and now extend this analysis to cellular loci. The authors use ChIP-sequencing to map CTCF binding at cellular sites in HPV positive cells and compare that in HFKs to find overall distributions are similar, however, with increases or decreases at specific loci. By integrating RNA-seq with ChIP seq data, the authors identify a series of sites where gene expression appears altered through differential CTCF binding. A cluster of 20 differentially bound CTCF sites is localized to a region of chromosome 11 that contains the CADM1 and ZBTB16 genes. CADM1 expression and protein levels are reduced in HPV positive cells while ZBTB16 expression is increased. The authors then characterize histone binding at these loci using ChIP-seq data and find reductions in H3K4me3 and H3K27Ac with increases in H3K27me3 around the CADM1 gene body. The CpG regions near the CADM1 TSS were hypomethylated in HPV positive cells as compared to the neighboring gene, APOA5.

This is an interesting study that examines the effects of differential CTCF binding on cellular gene expression during the establishment of high-risk HPV genomes. The study could benefit from a functional demonstration of the effect of CTCF binding on CAMD1 and/or ZBTB16 expression.

Reviewer #2: This is an interesting study where the authors have analyzed changes in cellular gene expression in response to episomal expression of HPV18. They show that the aberrations that they observe are likely driven mostly by E6 and E7 and that at least some of these result from epigenetic reprogramming due to changes in CTCF binding. This is one of the first studies that lend mechanistic credibility to the concept that HPV-expressing cells undergo epigenetic reprogramming. This paper also stands out by its high level of rigor, i.e., the analysis of several independently derived cell populations originating from different donors, the use of cutting-edge methods, and excellent statistical evaluation of the data. Overall, the authors' argument that some of the changes in gene expression are caused by differences in CTCF binding, which is NOT due to changes in DNA methylation, is well supported by the data that they present. The authors claim that the changes might be brought about by the increased levels of CTCF due to post-translational stabilization in HPV18-expressing cells, however, remains to be tested experimentally.

Reviewer #3: The manuscript by Campos-León and colleagues explores the impact on gene expression and chromatin by infection with HPV18 in normal human keratinocytes. In particular, they addressed whether the CTCF factor affected gene expression patterns. Using cells infected with HPV18 for at least 5 days, the authors performed whole transcriptome and identified differentially expressed genes in 6 unique primary infected cells. For two of the donors, they performed ChIP-seq for CTCF in paired normal and infected cells and identified a large number of peaks included a major shift for the infected cells to peaks away from gene bodies to intergenic and gene desert regions. Comparing changes in gene expression with changes in CTCF binding, they observed a small number of chromosomal regions that clustered CTCF binding changes near to gene expression changes. They focused on one region in Chr11 harboring the CADM1 gene that had decreased levels of expression, consistent with prior reports. Interestingly, CTCF binding was disrupted at the CADM1 enhancer. These effects were confirmed by chromatin conformation capture (4C-seq). Unexpectedly, the CADM1 promoter enhancer did not reveal an increase in CpG DNA hyper-methylation leading the authors to propose that the hypermethylation observed in prior studies of cervical cancer were a later event and occurred after the changes in histone modification with increased levels of H3K27me3.

In general, the manuscript was well written and the data performed at a high level of excellence.

**Part II – Major Issues: Key Experiments Required for Acceptance**

Reviewer #1: 1). What happens to CAMD1 or ZBTB16 expression if CTCF is knocked down? If CTCF is an essential gene, then short-term knockdowns with siRNAs may work. ZBTB16 can be assayed by RT-qPCR since the antibody is not very good.

2). Is there any looping between the HPV 18 CTCF site and CADM1?

3). While there was no difference in the total CTCF-bound peaks, were there novel CTCF binding sites within the HPV 18 + donors? Are the differentially bound CTCF peaks representative of novel sites or just altered binding (ie. Where the CTCF reads were at or below the input control level in one sample but significantly above that threshold in another)?

4). Is 100kb the standard/accepted genomic size to associate CTCF binding sites with gene expression? Since about 45% of your CTCF binding sites in both HPV 18 + and - cells were in gene bodies, proximal promoter, or distal promoter regions, why not use these binding sites (and any differential CTCF binding between HPV + vs -) to integrate the differential gene expression data?

5). In Table 1, are the differentially expressed genes representative of the average of the six donor samples (+ and -), or is the differential expression true for each donor? The consistent differential expression of ZBTB16 and CADM1 in each donor is striking. Clarification of whether some of the differentially expressed genes listed in Table 1 follow the same trend would support the mechanistic impact of differential CTCF binding.

6). Is the % CpG methylation of CADM1 TSS in the HPV 18 genome containing cells different than that seen in the HPV negative donors? This would seem to be a more direct comparison to determine hyper- versus hypomethylation in different cell lines.

Reviewer #2: is it possible to test whether the changes in CADM1 and ZBTB16 expression in HPV18-expressing cells can be reversed by manipulating CTCF expression (RNAi, CRISPRi)

Reviewer #3: The introduction cites repeatedly that the prior literature that HPV infection and cancer leads to increased levels of CTCF protein reported to be post-transcriptional. It would be useful if a specific comment about the levels of CTCF in the transcriptome of the normal and infected cells in the current experiments could be noted. In addition, it would be helpful to perform western blots of CTCF to evaluate if levels were increased in the infected cells. It is not clear if the current experiments were similar to the earlier reports from this lab (reference #30).

**Part III – Minor Issues: Editorial and Data Presentation Modifications**

Reviewer #1: (No Response)

Reviewer #2: 1.The font size in many of the figures should be increased to make sure that they remain legible.

2.In Figure 3B "Read counts" should be replaced with "CADM1 Read Counts" and ditto for "Fold Expression"

3.Reference 44 should receive a bit more discussion (lines 404 to 407)

4.Something is missing in the sentence "GSEA highlighted ..." (lines 410-411)

5.On line 416 delete "REF"

Reviewer #3: The redistribution of CTCF binding sites in the normal vs infected cells was reported in the text (lines 194-200) as a significant shift from genebody (40%) and promoter (14%) in normal cells toward intergenic and gene desert regions in infected cells. However, I find that the Venn diagrams in Figure 2C do not effectively illustrate this change as I am forced to visually estimate the size of the pie slice. Perhaps a bar graph or another type would be clearer.

The use of nanopore sequencing to identify DNA methylation was a very powerful approach to address the question of the impact of HPV18 infection on DNA methylation and if this contributed to changes in CTCF binding. However, it was not clear from the main text and methods if both the normal and infected cells were sequenced or was it just the infected cells from three donors. If just the infected cells were sequenced, how were the changes assessed on Figure 6? Furthermore, the discussion section that describes these effects (lines 454-463) is not clear. If the CADM1 region is hypermethylated in HPV-driven cancer, how does this relate to the regions that are described as hypomethylated in the manuscript?

PLOS authors have the option to publish the peer review history of their article (what does this mean? ). If published, this will include your full peer review and any attached files.

**Do you want your identity to be public for this peer review?** For information about this choice, including consent withdrawal, please see our Privacy Policy .

Reviewer #1: No

Reviewer #2: No

Reviewer #3: No

Figure Files:

Data Requirements:

Please note that, as a condition of publication, PLOS' data policy requires that you make available all data used to draw the conclusions outlined in your manuscript. Data must be deposited in an appropriate repository, included within the body of the manuscript, or uploaded as supporting information. This includes all numerical values that were used to generate graphs, histograms etc.. For an example see here: http://www.plosbiology.org/article/info%3Adoi%2F10.1371%2Fjournal.pbio.1001908#s5 .

Reproducibility:

References:

---

## [Editor Report · Decision Letter 1]

2 Dec 2024

Dear Professor Parish,

We are pleased to inform you that your manuscript 'Repression of CADM1 transcription by HPV type 18 is mediated by three-dimensional rearrangement of promoter-enhancer interactions' has been provisionally accepted for publication in PLOS Pathogens.

Best regards,

Cary A. Moody

Academic Editor

PLOS Pathogens

Patrick Hearing

Section Editor

PLOS Pathogens

Michael Malim

Editor-in-Chief

PLOS Pathogens

orcid.org/0000-0002-7699-2064
---

## [Editor Report · Acceptance letter]

Dear Professor Parish,

We are delighted to inform you that your manuscript, "Repression of CADM1 transcription by HPV type 18 is mediated by three-dimensional rearrangement of promoter-enhancer interactions," has been formally accepted for publication in PLOS Pathogens.

Best regards,

Sumita Bhaduri-McIntosh

Editor-in-Chief

PLOS Pathogens

orcid.org/0000-0003-2946-9497

Michael Malim

Editor-in-Chief

PLOS Pathogens

orcid.org/0000-0002-7699-2064